# Risk Factors for Brain Health in Agricultural Work: A Systematic Review

**DOI:** 10.3390/ijerph19063373

**Published:** 2022-03-13

**Authors:** Emily Terese Sturm, Colton Castro, Andrea Mendez-Colmenares, John Duffy, Agnieszka (Aga) Z. Burzynska, Lorann Stallones, Michael L. Thomas

**Affiliations:** 1Department of Psychology, Colorado State University, Fort Collins, CO 80523, USA; emily.sturm@colostate.edu (E.T.S.); andrea.mendez@colostate.edu (A.M.-C.); john.duffy@colostate.edu (J.D.); lorann.stallones@colostate.edu (L.S.); 2Department of Biomedical Sciences, Colorado State University, Fort Collins, CO 80523, USA; castroc@colostate.edu; 3Molecular, Cellular and Integrative Neurosciences Program, Colorado State University, Fort Collins, CO 80523, USA; aga.burzynska@colostate.edu; 4Department of Human Development and Family Sciences, Colorado State University, Fort Collins, CO 80523, USA

**Keywords:** agriculture, brain health, risk factors, pesticides, vulnerable populations

## Abstract

Certain exposures related to agricultural work have been associated with neurological disorders. To date, few studies have included brain health measurements to link specific risk factors with possible neural mechanisms. Moreover, a synthesis of agricultural risk factors associated with poorer brain health outcomes is missing. In this systematic review, we identified 106 articles using keywords related to agriculture, occupational exposure, and the brain. We identified seven major risk factors: non-specific factors that are associated with agricultural work itself, toluene, pesticides, heavy metal or dust exposure, work with farm animals, and nicotine exposure from plants. Of these, pesticides are the most highly studied. The majority of qualifying studies were epidemiological studies. Nigral striatal regions were the most well studied brain area impacted. Of the three human neuroimaging studies we found, two focused on functional networks and the third focused on gray matter. We identified two major directions for future studies that will help inform preventative strategies for brain health in vulnerable agricultural workers: (1) the effects of moderators such as type of work, sex, migrant status, race, and age; and (2) more comprehensive brain imaging studies, both observational and experimental, involving several imaging techniques.

## 1. Introduction

Agriculture represents a critically important sector of the workforce. Indeed, almost 27% of the global workforce is currently employed in agriculture [1]. In the US, approximately 2 million full-time workers were engaged in agriculture in 2018 [2]. Agriculture has more workers over the age of 55 than any other industry [3], and workers in agriculture are at increased risk for psychiatric and neurological disorders compared to other professions [4]. Despite these concerns, agricultural workers are underrepresented in brain related research [5,6,7,8].

People working in agriculture have some of the highest rates of brain cancer and Parkinson’s disease [9,10,11], 46% greater odds of having dementia [12], and increased rates of anxiety, depression, and suicide [13,14,15,16]. These risks may be associated with central nervous system dysfunction; for example, agriculturalists have been found to experience a higher risk of impaired neurobehavioral, neuromotor, or neurocognitive performance [17]. However, the mechanisms of these deficits are poorly understood. This study aims to identify gaps in knowledge and areas for future research.

Understanding life experiences of the individual at work is a key element to understanding their brain health over time. However, the study of risk factors that are associated with agricultural work is complex. Even the definition of agricultural work can be elusive. “Agricultural” is an umbrella term that encompasses many distinct occupational activities. This includes cultivation and tillage of soil, dairy production, cultivation, growing, and harvesting of crops, raising livestock, raising furbearing animals, raising poultry, preparing agricultural products, packinghouse work, nursery and landscaping operations [18]. Moreover, within these categories work is further divided by position of employment (e.g., owners, management, graders and sorters, equipment operators, truck drivers, etc.). Given this heterogeneity, it is reasonable to expect that agricultural risks for poor brain health are highly variable between settings and groups. Indeed, research suggests that that agricultural risks are not consistent across different geographic areas and types of agricultural work [19]. Moreover, biological sex, and job position may moderate the impact of these risk factors in the brain [10,20,21].

Variability in the types of agricultural jobs performed, and who performs them, implies that a comprehensive understanding of the topic is required to recommend interventions. Certain risks—such as exposure to pesticides—are common and relatively well studied. Although several reviews have been conducted in this area, they have not focused on brain outcomes. Instead, articles have reviewed mental health in agricultural work without an explicit focus on brain health [22,23], chemical exposures encountered in agriculture and Parkinson’s Disease [24,25,26,27,28,29] or brain cancer [9,30,31,32,33,34]. Three reviews investigate the influence of pesticide exposure on neurological impairment via neuropsychological testing but lack methods that can explain the brain-based mechanisms [35,36,37]. There is a wealth of literature on neurotoxicological outcomes of acute and short-term exposure to pesticides in laboratory animals, but there are relatively few studies of long-term exposure. Many reports in the literature describing ‘chronic’ exposures to pesticides are, in fact, as short as five days and rarely longer than three months. Furthermore, routes of administration range from subcutaneous to dietary. Doses used in many of the studies produce signs of acute or overt toxicity. In contrast, human symptoms have been reported following exposures that are prolonged and often without obvious toxic effects.

A survey of the literature was conducted to identify rodent studies with neurobehavioral and neurophysiological endpoints of pesticide exposures lasting 30 days or longer [38]. This survey indicated that the majority of studies concentrate on cholinesterase inhibitors (organophosphorus and carbamate insecticides). Various neuromotor, cholinergic, physiological, affective, and cognitive disorders were reported at doses producing cholinesterase inhibition; however, there were fewer effects at non-inhibiting doses. In many studies, the changes were subtle, which may correspond to the nonspecific changes in psychomotor and cognitive function reported in humans. It appears, then, that the data from animal and human pesticide exposures are generally comparable, but the specific outcomes are influenced by many experimental differences. Future research should concentrate on analogous exposures and outcomes to facilitate interpretation [38]. Without a systematic review that integrates findings from these diverse literatures it is challenging to know how best to study and ultimately intervene in this area.

Given the importance of occupational exposures to brain health, understanding work risks can help researchers advise how best to structure work environments to optimize brain health over time [39,40]. For example, understanding how experiences associated with work affect the brain could lead to interventions that are designed to promote health among the industry’s workers [41]. This could be particularly important to agriculturalists, due to the increasing number of workers over the age of 55. Neurological disorders commonly associated with age, such as Parkinson’s disease and dementia, are more prevalent among agriculturalists, therefore interventions could be especially helpful in promoting healthy brain aging as this industries workforce continues to increasingly consist of older adults.

To constrain our review, we focus on findings that are “brain-based”, operationally defined as studies that examine the effects of agricultural risks on known brain regions. This includes clinical diagnosis by trained research professionals of diseases such as Parkinson’s disease, Alzheimer’s disease, and brain cancer. Clinical diagnosis of depression, neurobehavioral deficits, and cognitive deficits were not included due to our focus on outcomes strongly linked to specific brain abnormalities. This is because the impact of agricultural work on the brain is often observed using indirect methods, such as self-report questionnaires or occupational census data instead of direct measures such as neuroimaging and histology. Additionally, we constrained our review to articles that specifically concerned agricultural populations. Although additional literature exists on risks like heavy metal or pesticide exposure identified in other contexts like welding or pesticide production, our aim was to provide a pragmatic understanding of how well these factors have been studied specifically within agricultural contexts.

The primary aim of this article is to systematically review the literature to identify both known and putative risk factors for poor brain health that are associated with agricultural work. Consistent with this, we aimed to identify sparse areas of the literature that need additional research. Finally, we aimed to identify important moderators of the relationship between risk factor and brain outcomes.

## 2. Methods

We conducted a systematic search of articles in MEDLINE, AGRICOLA and CABabstracts on 30 November 2021. MEDLINE is a bibliographic database that contains references to journal articles in life sciences with a focus on biomedicine. AgriCOLA contains bibliographic records from the U.S. department of Agriculture’s National Agricultural Library. CABabstracts covers the significant research and development literature in the applied life sciences, including agriculture, forestry, human nutrition, veterinary medicine, and the environment. We searched for published studies, not reviews, not case studies, and not preprints, in peer-reviewed journals, applying no filters on publication year or language. To identify results related to agriculturalists, the following search terms were used: agri*, farm*, ranch*, dairy, horticult*, plantation, livestock. Additionally, to identify results related to occupations, the following search terms were used: work, occupation*, job, career. Finally, in order to identify results related to brain-related outcomes, the following search terms were used: psychia*, dementia, Parkinson’s, neurocogniti*, memory, mental, brain, white matter, gray matter, nervous system, cortical, subcortical, neuroimage*, neuro*, magnetic resonance imaging, functional near-infrared spectroscopy, positron emission tomography, diffusion-weighted imaging, transcranial magnetic stimulation, transcranial direct current stimulation, cogniti*, MRI, fMRI, CT, PET, FNIRS, DWI, DTI, TDCS, TMS, gut brain axis, and head injury.

In order to select articles that fit inclusion criteria, two authors independently screened the title, abstract and, where appropriate, full text of identified citations. Disagreements were resolved by consensus (including discussion with the senior author). Our inclusion criteria were:

1. Identified a risk. Risk factors were defined as any exposure that was investigated for a relationship to poor brain health. If articles examined multiple risk factors relating to the brain, they were included in each of those categories. Thus, some articles appear twice in the results.

2. Explicitly related to agricultural work in the adult population. Developmental studies were not considered due to the existence of other reviews and meta-analyses that focus on neurodevelopment in relation to agricultural risk factors [17,42,43,44,45,46,47]. As noted in the introduction, agricultural work is an umbrella term that includes many distinct job activities. This includes cultivation and tillage of soil, dairying, cultivation, growing, and harvesting of crops, raising of livestock, raising of furbearing animals, raising of poultry, preparation and delivery of agricultural products, packinghouse work, and nursery and landscaping operations [18]. All these categories were considered agricultural work for the purpose of this review. Animal studies were included if they discussed implications for agriculture. For example, a study using mouse microglial cells to investigate the neuroinflammatory effects of organic dust was included because the dust used in the study was collected from a farm where the risk of inhalation is present for agricultural workers thus making the research directly relevant to occupational risk factors [48]. However, multiple animal studies were excluded, despite focusing on well known risk factors in agriculture, like pesticide exposure, due to a lack of explanation of how their methods specifically relate to agricultural work. 3. Contained either a direct measure of the brain and/or a clinical measure of a disease or disorder with a known link to a specific brain abnormality. Direct measures of the brain include imaging, histology, or death record indicating specific brain-based cause of death (e.g., brain tumor) and clinical examination. Animal studies were also included if they contained a measure of the central nervous system (e.g., histological examination of neurons or glia) with an explicit agricultural context. Animal studies were only included if the methods used were intended to be translational to an agricultural occupation. For clinical disease or disorder, we sought the brain-basis of the outcomes related to agricultural work; thus, a measure of the brain or knowledge of the specific brain area affected was required for inclusion. For example, Parkinson’s disease was included but depression was not because there is a very specific brain area associated with Parkinson’s while the brain-specific basis of depression is more ambiguous. Studies that only indirectly measured brain function through neurocognitive testing or self-report were not included.

A flow diagram of the study selection is shown in Figure 1. Our initial search resulted in 1501 unique records. Two research associates separately screened the titles and abstracts of articles into those that fit the criteria (170) and those that did not (974). Then, one research assistant further sorted the articles, ensuring each article met all criteria (106). The resulting 106 articles were then sorted according to risk factor(s) studied in the article. If an article identified more than one risk factor, they appear in each risk factor category in the results. Next, brain areas associated with those risk factors were identified for each article. Finally, moderating factors and type of research were identified and synthesized in the discussion.

## 3. Results

We identified seven unique risk factors shown in Table 1. We identified 23 brain areas associated with these risk factors shown in Table 2. In order to identify areas with sparce research, Table 2 shows the risk factors and relevance to the brain with darker gray representing a higher number of articles in that intersection, lighter gray representing less articles in that intersection, and white representing no articles at that intersection.

### 3.1. Non-Specifical Factors Associated with Agricultural Work

Studies, all human and mostly epidemiological, have considered agricultural work as a risk factor without necessarily identifying specific attributes of the job that are related to outcomes. Of the 104 articles included in the final review, 33 fell into this category.

In terms of methodology, case-control studies were common, especially when investigating brain cancer, with many of these relying on diagnoses obtained from death certificates. In terms of location, the majority were from the US or France see Figure 2. The geographic regions examine ranged from as small as Detroit to as large as eight collaborating study centers across the world which suggests that agricultural workers around the world are considered vulnerable to certain exposures that impact brain health with little indication of differences based on geography.

Not all studies agreed on the association between agricultural work and increased risk of brain disease, specifically in the two best studied areas: brain cancer and Parkinson’s disease. However, possible sex-based or age-based differences, and lack of measurement of specific job-related activities, such as pesticide application, were identified as possible reasons for lack of association in some studies [55,56,59,68]. Several of these articles suggested that there are sex-based differences [21,51,61,66]. In other words, sex and age are a moderator of the relationship between agricultural occupation and brain health and should be considered during analysis.

Studies differed in their measurements and definitions for agricultural work/occupations. For example, some studies relied on occupation at time of death which may represent the highest level of occupation or could represent a change in occupation due to semi-retirement. Other studies included agricultural occupation as a factor if the person worked an agricultural job for more than 6 months, and still others conducted detailed interviews with participants to find out if agriculture was the occupation they worked for majority of their adult life. Differences in agricultural activities may result in very different exposures to the putative agent of disease, thus measuring this risk factor is both important and complex.

#### 3.1.1. Dementia

Of the 33 non-specific risk articles, two found an association between agricultural work and increased risk of dementia. One study from France—which defined the agricultural occupation under investigation as “farming”—found no association between occupation and dementia, in general, except that dementia with Parkinsonism seemed to be increased, particularly among women, in farmers compared with other occupations [21]. A study from the US considered all jobs in agriculture and found greater odds of having dementia for those in the agricultural sector [12].

#### 3.1.2. Brain Cancer

Of the 33 non-specific risk articles, 11 examined the relationship between agricultural occupations and brain cancer. Most found an association between increased rates of brain cancer; however, one study, which collected detailed job histories, did not [55]. Interestingly, studies found differences in brain cancer rates by agricultural job type [61,67], race [61,64], socio-economic status [61], sex [61,66], and pesticide exposure [49,51,57].

A study from New Zealand found that dairy farmers, sheep farmers, and livestock workers had the highest odds ratios for brain cancer compared to other job types, including other job types within agriculture [61]. Other studies relied mostly on broad classification of agricultural work.

Two of the studies focused on brain cancer found differences by race. A US study that used over 4.5 million death certificates found that agricultural exposures were associated with a significant increase in central nervous system cancers among White women and men, but not Black women and men [64]. Notably, the authors did not distinguish between different types of central nervous system cancer but did mention that excluding cancers other than the brain did not modify the risk estimates [64]. A study from New Zealand found that Māori had a higher incidence rate for brain cancer than non-Māori [61]. Another study found that education and race (White vs. non-White) interacted with brain cancer rates [49].

Socioeconomic status was examined in a study from New Zealand. Specifically, the authors examined socioeconomic status and its relation to brain cancer incidence among various occupations, including agricultural workers [61]. Results indicated that higher socio-economic status was associated with higher incidence of brain cancer. Notably, this result was only in males.

Of the nine studies that found a relationship between agricultural occupations and brain cancer, one studied only women, one studied only men, and one found the association was only present for women [49,51,66]. Sex was investigated in three of the studies about brain cancer and agricultural work. In New Zealand, women had generally lower incidences per 100,000 compared to men from 1953–1988 [61]. In the US, agricultural occupations were associated with increased rates of brain cancer in women but not men [66]. Internationally, women within the agricultural occupation category had a reduced risk of glioma [57].

#### 3.1.3. Parkinson’s Disease

Of the 33 non-specific risk articles, 22 examined the relationship between agricultural work and Parkinson’s disease or Parkinsonism but they did not all agree on the presence of an association. For example, in a population-based cohort study from Greece, researchers found significantly higher risk for Parkinson’s disease among farmers; however, the association became non-significant after adjusting for age [56]. Two studies from France found that farming, or having a primary job in agriculture, was not associated with Parkinson’s disease [25,140]. Similarly, no association between farming and Parkinson’s disease was found in a survey conducted in Italy [59].

However, four separate studies in France found higher incidences of Parkinson’s disease among farmers as well as agricultural workers more broadly defined [21,50,53,63]. This risk was especially high among women [21]. Similar associations between agricultural work and Parkinson’s disease have been reported in samples from China [50], Norway [62], Africa [48], Italy [60], Pakistan [63], and the US [65]. Notably, a study from Africa found a link between agricultural work and Parkinson’s disease even after controlling for age, sex, family history of Parkinson’s disease, head trauma, and smokeless tobacco (snuff) use [48]. A study from Italy found that controlling for smoking eliminated the association between farming occupations and Parkinson’s disease as smoking is a putative protective factor against Parkinson’s disease [60].

The reasons for inconsistent findings linking agricultural work to Parkinson’s disease are unclear. Some authors have pointed to issues such as misclassification and inadequate response rates as possible explanations for the non-association [59,140]. However, perhaps a more likely explanation concerns the heterogeneity of agricultural work itself. For example, a large longitudinal cohort study from Denmark examined risks of specific types of agricultural work [52]. The authors found an increased risk of Parkinson’s disease among self-employed farmers and a consistent but insignificant pattern of high risks among other occupations involving pesticides (see below for further results on pesticide exposure). In other words, the explanation for inconsistent associations between agricultural work and Parkinson’s disease appears to be that the specific type of agricultural work, as well as setting, sex, and age, matter.

### 3.2. Airborne Toluene

Toluene is an aromatic hydrocarbon present in gasoline. Agriculture was identified as a potential source of exposure to this highly volatile contaminant which can cross the blood-brain barrier and has been investigated in the past for its association with central nervous system damage [141]. A study using *C. elegans* as a model organism found that swimming movements were reduced, and GABAergic and cholinergic neurons had dose-dependent changes in fluorescence intensity and morphology [77]. Although this study was conducted with a model organism, it points to important neuronal changes that could also occur in the human brain.

### 3.3. Dust

While dust has been previously explored for its relation to lung inflammation, new research suggests an important link between dust exposure and the brain. We identified one study that specifically focused on airborne organic dust with a direct measure of the brain. In study conducted in the US, Massey and colleagues found that organic dust induced neuroinflammation in micro glia and attenuated HMGB1-advanced glycation end products (RAGE) activation [78]. Importantly, the authors investigated the type of dust that farmers working with pigs, dairy, and poultry commonly encounter in the absence of respiratory protection [142]. This kind of dust contains particulates such as lipopolysaccharides, peptidoglycan, and microbial DNA, all of which can cause neuroinflammation. Thus, it is important to consider neural implications, not just lung inflammation when studying dust exposure. Notably, although not included in this systematic review, previous research in Australia has found that farmers are routinely exposed to unsafe levels of dust [142] and a study from Korea linked dust exposure with poorer mental health [143]. Thus, there is reason to pursue more mechanistic approaches to understand the impact of dust exposure on the brain.

### 3.4. Farm Animals

Exposure to farm animals is a risk factor that is closely associated with some but not all agricultural tasks. Exposure to farm animals can occur both for farmers who raise animals and agricultural workers who are responsible for slaughter. Several articles suggested that animal exposures can have brain-based impacts on agricultural workers. Of the five articles that fell into this category, we identified two studies that found links to multiple sclerosis [81,82], one that found a negative association with Parkinson’s disease [83], one that found a positive association with brain cancer [80], and one that found no association with brain cancer [79].

Two case-control studies had differing results on the link between exposure to farm animals and incidence of multiple sclerosis. One study in Australia found an increased risk among women with 10 or more years of exposure to livestock increased risk for women with 6 or more years of farming experience [82]. However, a separate study in Australia found no association [81]. These conflicting results could be related to sex differences which make the relationship between animal exposure and multiple sclerosis especially difficult to understand [81,82].

Two studies also had differing results on the link between exposure to farm animals and brain cancer [79,80]. While a study that took place in the US found an increased risk of brain cancer associated with killing chickens, an international study did not find an association between exposure to farm animals and brain cancer [79,80]. Additionally, a study in Finland found that animal ownership (including cows, sheep, pigs and chickens) was associated with lower rates of Parkinson’s disease but due to small sample size could not determine if animal ownership was a protective factor or merely a proxy for other lifestyle factors and they did not differentiate between agricultural or companion animal ownership of the animals so results may be confounded by pet-owners [83].

### 3.5. Heavy Metals

Exposure to heavy metals can happen in myriad ways in agriculture. Six articles—one epidemiological, one using brain imaging on humans, and four animal studies—investigated the impacts of manganese or heavy metal exposure on the brain [84,85,86,88,89]. Manganese was the most commonly studied heavy metal exposure. Manganese is an ingredient in some fungicides used in agriculture and can also be emitted into the air by ferromanganese plants and welding fumes. Other metals, such as lanthanide, arsenic, nickel, zinc and copper are also commonly encountered in agricultural work [88,89,144]. A study from Taiwan found that cognitive dysfunction, cerebellar ataxia, and Parkinsonism were possible outcomes of heavy metal intoxication in agricultural workers [86]. A human brain imaging study from Costa Rica measured manganese concentration in farmworkers’ toenails and hair, and then used functional near-infrared spectroscopy—a type of functional brain measure—to explore brain activity related to exposure. This study did not find an association between manganese concentration and brain activity. However, this study was based on a small sample (*n* = 48), and was therefore low-powered to detect significant effects [84]. Additional possible reasons for the null finding in the study from Costa Rica could be that farm workers did not have as severely elevated levels of exposures as welders, differences in tasks used, and variations in procedures used to prepare biological samples [84]. Indeed, animal research has shown strong evidence for the risk of manganese on brain health. For example, an animal study examining drosophila who ingested manganese suggested that changes in cholinergic and dopamine levels were caused by manganese [85]. The authors also investigated whey protein isolate as a protecting factor against these effects as part of their aim to improve outcomes from occupational exposure to heavy metals [85]. Another animal study using rats found that concentrations of manganese increased in the striatum, altering noradrenaline, dopamine, 5-hydroxytrytamine, glutamate, and acetylcholinesterase concentrations in that area [87]. Importantly, this study observed rats after low dose, subchronic pesticide exposure far lower than what would usually be considered acute poisoning and more similar to an agricultural worker who may handle pesticides safely enough to experience only low dose exposure [87].

### 3.6. Nicotine Exposure

Farmers working with tobacco are likely to be exposed to elevated levels of nicotine in their bodies due to handling tobacco plants on the job. One case-control magnetic resonance imaging (MRI) study among Latino farmworkers found that farmworkers had lower gray matter signal in areas including the right ventrolateral prefrontal cortex and right dorsolateral cortex, and higher gray matter signal in the putamen and cerebellum [90]. When urinary cotinine levels—a measure of nicotine exposure—were included in the analysis, the areas of increased activity in the prefrontal cortex nearly doubled in size. Smoking history was included as a covariate and did not significantly alter the results [90]. A functional magnetic resonance imaging study on the same population of Latino farmworkers in the US to investigate functional connectivity found that urinary cotinine levels were associated with a difference in modularity of functional brain networks between farmworkers and non-farmworkers [91].

### 3.7. Pesticides

Agricultural workers experience increased rates of exposure to pesticides due to pesticide mixing, loading, application, living in areas where pesticides are commonly applied, accidental acute exposures, and chronic low-level exposures [145]. The consequences of occupational pesticide exposure on neurodevelopment, neurocognition, and certain brain-health issues like brain cancer have been previously reviewed at length [9,29,32,35,44,46,146,147,148]. This review uniquely examined articles relating to brain health in agricultural workers exposed to pesticides.

Of the 104 total journal articles reviewed here, 57 investigated the relationship between pesticide exposures in agricultural work and known areas of the brain. Of the 57 articles concerning pesticide exposure, 20 considered all pesticides without mentioning subcategories, nine focused on insecticides [96,98,100,104,105,109,113,116,149], six focused on organophosphate pesticides [92,93,95,103,112,117,123,139], three focused on herbicides [111,133,136], and the rest focused on either a specific pesticide (e.g., cypermethrin) [134], or a type of pesticide defined by the type of agriculture where it’s used (e.g., pesticides used on vineyards) [137]. Because some reviews exist on certain specific types of pesticides, we included all chemicals used to eliminate insects, fungi, and unwanted plants in our definition of pesticides and do not separate them into subgroups below [24,27,42,150,151]. Notably, Parron and colleagues separated pesticides into categories like insecticides, fungicides, herbicides, plant growth regulators and other pesticides [118]. They found that areas of Spain with highest pesticide use also had the highest rates of Alzheimer’s disease, Parkinson’s disease, suicide, and neuropsychiatric disorders; however, the type of pesticide did not affect this relationship [118]. Yet, some studies have found that the presence of active ingredients such as paraquat or maneb strengthened associations with brain damage [99,119,152]. Paraquat and maneb are common ingredients in herbicides and fungicides respectively and have been studied for their highly toxic properties [153,154]. Below, we discuss pesticides, not divided by subtype of pesticide or type of exposure but by brain area impacted.

Pesticides can be examined in terms of acute or chronic exposure. However, very few articles have explored the combination of both, which is likely a more accurate reflection of agricultural work. Participants in studies focused on pesticides tended to be older adults possibly due to the ability to study chronic exposure, the possibility that pesticides exacerbate age-related brain illness like Parkinson’s, and changes in pesticide regulations that removed some pesticides from the market.

#### 3.7.1. Brain Cancer

Dividing farms by crop type, animal type, and potential for exposure to pesticides revealed that pesticides significantly impacted risk for brain cancer [98]. A study in France found that agricultural workers overall did not have higher rates of brain cancer, but that working in agriculture for over 10 years and pesticide exposure were strongly associated with brain cancer suggesting that pesticide exposure and not agricultural work itself may be the main risk factor [97].

#### 3.7.2. Parkinson’s Disease

The studies that focused on pesticide exposure and Parkinson’s were either case-control, cross-sectional, cohort, or animal studies. Only one study found no association between pesticide exposure and Parkinson’s disease [83]. The authors of this study suggested that lack of association could be related to the small number of participants using paraquat pesticides and the historically low levels of pesticides being used in Finland where the study was conducted [83]. When considering pesticide exposure, multiple studies pointed to an important genetic influence; specifically, some individuals are more likely to experience Parkinsonism due to genetic differences in genes that encode dopamine receptors [116,117,121]. There appear to be important sex-based differences with men at greater risk than women for Parkinson’s disease after occupational exposure to pesticides [120].

#### 3.7.3. Other Brain Related Impacts

Animal studies constituted 18 of the 57 articles regarding agricultural occupational exposure to pesticides and brain health. Importantly, there are many more animal studies about the negative effects of pesticides on the brain but only the articles included in this review framed their work in context of agricultural work. An article investigating deltamethrin tested acute exposure on rats similar to the kind of acute exposure that could accidentally happen during agricultural work [109]. The authors found that acute deltamethrin exposure causes motor and cognitive impairments in rats and may be related to disruption of the dopaminergic pathway, specifically reduction of tyrosine hydroxylase immunoreactivity in the substantia nigra pars compacta, ventral tegmental area and dorsal striatum [109]. Notably, whereas this study used intracerebroventricular administration of the insecticide, others have used oral ingestion [92,96,103,113]. One study used MRI and texture analysis on mouse brains. The authors found that the hippocampus and somatosensorial cortex were impacted by an intraperitoneal, chronic, low-dose, injection of glufosinate ammonium herbicide [111]. Another study using rats investigated deltamethrin and other type 2 pyrethroids that are commonly used in agriculture including cyfluthrin and cyhalthrin [105]. The authors investigated acute exposure by injection and found a depleting effect on serotonin in midbrain and striatum areas [105]. Yet another animal study found that organophosphorus pesticides were associated with disrupted tubulin polymerization which could be a mechanism of neurotoxicity [112].

Rats have also been used as a model organism when investigating imidacloprid exposure [113]. Imidacloprid is a neonicotinoid insecticide, used on crops around the world that acts selectively as an agonist on insect nicotinic acetylcholine receptors. Importantly while other studies often examine acute exposure, one study used a range of low doses over a 28-day period. The results of imidacloprid exposure included damaged DNA and oxidative stress in the brains of the rats [113]. In comparison, another study exposed rats for 90 days to a common insecticide (acetamiprid) and found enhanced oxidative stress in brain mitochondria as well as increased permeability and swelling in the brain mitochondria of chronically exposed rats [96].

One unique study investigated the protective properties of Crataegus oxyacantha extract on rats that were administered low doses orally of deltamethrin and chlorpyrifos [104]. The authors found abnormal cellular morphology accompanied by severe vacuolation and necrosis of neurons observed in cerebral cortex of the combination of deltamethrin and chlorpyrifos (DCF) induced neurotoxicity. They also found that Crataegus oxyacantha extract and vitamins C and E attenuated this effect [104]. This is important because the doses they use correspond to very low doses in humans, below the acceptable daily limit. Occupational exposure could certainly be at this level if not much greater.

The human studies that investigated brain-related impacts other than Parkinson’s disease and brain cancer found that pesticides are linked to more modular functional networks, reduced gray matter signal, pineal gland disfunction, and sensory/motor conduction slowing [90,91,133,135]. Importantly, one of these studies linked pesticide exposure to changes in pineal gland activity and suggested a synergistic effect with aluminum exposure [133]. Still, one study found only a weak effect of chronic low dose exposure to organophosphate pesticides and separating the effects of exposure intensity and duration revealed a higher presence of neurological symptoms in sheep dippers who handled pesticides than agricultural workers that did not handle pesticides [139].

## 4. Discussion

Although agrarian societies are becoming less common, the role of farming, ranching, and other agricultural work is no less important today than hundreds of years ago [155,156,157]. Unfortunately, the nature of this work exposes agricultural workers to increased risks. As is clear from this systematic review, agricultural work is associated with myriad risks for poor brain health. We identified seven risk factors: non-specific factors associated with agricultural work, toluene, dust, farm animals, heavy metals, nicotine exposure, and pesticides. Moreover, we found evidence that agriculture work increases risk for changes in brain function, as reflected by changes in dopaminergic pathways, and cholinergic pathways and increases risk for disease, such as Parkinson’s disease and brain cancer [65,81,90,93,137,158]. Inconsistencies in the literature are likely due investigators failing to account for moderators such as delineating specific work environments (e.g., use of pesticides or job subtype), while other times there may complex synergies among risks (e.g., pesticide exposure and heavy metal exposure) [133]. Although our review highlights certain common work safety concerns for brain health (e.g., pesticide exposure), some areas of risk in agricultural work (e.g., animal exposure, vibration, heavy metal exposure) need further investigation.

### 4.1. Moderators

Articles that investigate agricultural work in general are useful for pointing research in the direction of brain areas that could be injured by occupational exposures in agriculture. However, methodological difficulties like characterizing agricultural work, genetic, and environmental differences must be addressed to understand the risks that workers endure. Additionally, mixed exposures are rarely studied but represent an important line of research in brain health. These important moderators of the impact of agricultural work represent an area where further study is needed.

Characterizations of the job subtype and duration of time spent in agriculture are often overlooked, especially in epidemiological studies which are the most common in this area of research. Studies that did seek further information about job subtype often found differences between job subtypes [55,61,65,99]. Estimating time spent in the agricultural occupation is another important consideration [97]. Multiple studies found that the number of years spent in the profession moderated the relationship between agricultural work and brain health, possibly enhanced by exposure to pesticides [97,122]. The need to define agricultural work and collect meaningful measures of duration of time in that occupation and type of job within agriculture were prominent lessons gained from this review.

Population subgroups such as women, migrants, and racial/ethnic minorities and low socioeconomic status individuals may be uniquely vulnerable to the brain-based outcomes of agricultural exposures [21,57,61,66]. Sex-specific effects cannot be determined from much of the research included in this review due to the recruitment of mostly male participants. While the industry is overwhelming staffed by male workers (68% in 2016), females are increasingly working in agriculture (down from 72% men in 2014) [159]. Furthermore, while principal operators are more likely to be male, secondary operators are more likely to be female [160]. Studies that did include females found increased rates of Parkinson’s disease among male farmers and increased rates of dementia and brain cancer among women farmers [21,53,67,74,120]. Additionally, females who perform “off-farm” labor may also be exposed to many of the same risks of agricultural work even though they are not counted among the employed agricultural population [161]. Importantly, men may be more likely to experience injury and pesticide exposure on the job [162]. It is likely that these findings, overall, reflect some combination of differences in sex defined as a biological variable as well as also socio-cultural factors related to gender norms in the work environment. As gender norms change, risk factors and affected groups will also likely change with them. While some articles not included in this review measured the rate of exposure for migrant workers and found migrant workers to have higher rates of exposure to pesticides than non-migrants, these studies left a gap of knowledge by failing to measure brain-based outcomes [7,21]. Given that migrant workers make up the majority of the agricultural workers, including migrant workers in agricultural research will help fill critical knowledge gaps for an increasingly important segment of agriculturalists. Race and ethnicity are studied in research about agriculture and mental health but very few studies take the extra step—albeit a costly one—of investigating impacts on brain structure and function. One study that investigated trends in socio-economic status associated with Parkinson’s disease rates found that farmers had increased risk, but socio-economic status had a relatively small effect on the likelihood of hospitalization for Parkinson’s disease [75]. Surprisingly, another study found that higher socio-economic status was linked with increased rates of brain cancer, although the same study did not find elevated risks for farmers like most other epidemiological studies [67]. It is possible that using death certificates as a measure of occupation limits the ability to fully explore these effects since the occupation coded is “usual” occupation and may miss occupational exposures from other than usual occupation. Methodological challenges also impact the ability to study socio-economic status. For example, brain-imaging equipment is typically located in more urban areas where it is difficult to recruit agriculturalists for research. Additionally, farmers’ lifestyles can often leave them with little flexibility in free-time and scheduling that would be needed to participate in a lengthy research study [163]. Future research should make stronger efforts to take socio-economic status, education, and cultural barriers to research participation into consideration [164,165].

Despite the fact that agriculturalists likely face mixed exposures to the risk factors reviewed here, mixed exposures are rarely studied. In an exception, one recent article found that the combined effects of corticosterone, an indicator of physiological stress, and chlorpyrifos, a common pesticide, was related to neuroinflammation and effects on neural signaling. More studies like this are needed if we hope to fully understand risks for brain health in this population. Overall, working in an agricultural occupation is a general risk factor for brain health. Unfortunately, many of these studies are missing information on job subtype and time of exposure. Indeed, important differences are blurred when they could be better assessed with more in-depth measurement of sex, migrant status, race, and ethnicity. There is a gap in research when it comes to more thorough measures of agricultural exposures and brain outcomes.

### 4.2. Pesticide and Chemical Exposures

Pesticides were the most prominent topic in agricultural research as identified by this review. Though many of these studies focused on pesticides’ association with brain cancer, others found evidence for effects of occupational pesticide exposure in the hippocampus, amygdala, motor and sensory cortexes, and the pineal gland. Neuropathology associated with pesticide exposure seems to have many expressions. These effects range from reduced antioxidant enzyme activity to impacts on functional brain networks [91,94]. Even genetic differences play a role in the rate at which agricultural workers experience adverse outcomes of pesticides, and thus should be carefully considered when searching for trends in brain health among agricultural workers [116,121,123,127]. Additionally, the nigrostriatal dopaminergic system was the most studied brain region due to its well-described vulnerability to pesticides and important implications with Parkinson’s disease and other movement disorders. Expanding understanding of other brain areas impacted by pesticides will help aid in future efforts to identify early signs of disease and disorder as well as contribute to protective measures.

Within the massive quantity of human and animal literature about pesticides, a modest number of studies were included in this review due to the lack of focus on agricultural occupational exposure and the need for more neuronal measures of pesticide impacts specifically in an agricultural context. Importantly, mental health as it relates to the brain and pesticide exposure may be intertwined in ways not yet investigated. In one study from India, 67% of organophosphate poisoning cases admitted to a hospital were suicide attempts [166]. Some of the reasons for suicide attempt included work-related stress and an additional 17% were occupational exposures [166]. Another study found that a biomarker of stress, corticosterone, increased neuroinflammation in rat brains, suggesting an important role of workplace stress [107]. These studies point to the classic chicken and egg situation because they cannot unravel the possibility for pesticide exposure to impact mental health which then impacts pesticide safety behaviors and effects of pesticides in the brain. Future studies could focus on localized brain effects to larger brain network and brain network interaction effects as they relate to behavioral outcomes.

### 4.3. Other Risk Factors

While pesticides get the most attention, other physical exposures such as airborne toluene, nicotine exposure, heavy metals, animals, and dust should be investigated further for consequences in the brain. The associated complications include grey matter reduction, damage to GABAergic, cholinergic and dopaminergic pathways [77,84,86,90,167].

Occupational exposure to organic solvents like toluene has been studied in workers from paint factories and semiconductor industries but rarely in the context of agricultural work, even though there is evidence that agricultural workers are routinely exposed to organic solvents from farm machinery maintenance and repair [168,169,170]. Sub-chronic exposures to toluene at sub-toxic levels can also alter catecholamine synthesis rates in the brain leading to sympathetic nervous system dysfunction, as demonstrated in animal models [171]. A study using *C. elegans* as a model organism showed neurochemical alterations in cholinergic and GABAergic systems [77]. The array of neurochemical dysfunction associated with organic solvents exposure manifests as atrophy and memory impairment in the long-term. Animal studies using rats and Drosophila have shown increased dopaminergic degeneration after manganese exposure, which could explain the extrapyramidal symptoms observed in humans [85,87].

Moreover, there is emerging evidence that the gut-brain axis is negatively impacted by stress and chemicals to which agriculturalists are routinely exposed [172,173]. Still, there is a need to translate these findings to human populations. Animal models cannot fully reflect the phenotype of human disease, but they have provided information on the neurobiological mechanisms of brain disease in metal intoxications [174]. Similarly, long-term exposure to high concentrations of heavy metals such as manganese has been associated with Parkinsonism, with the most affected brain regions within the basal ganglia [86].

Evidence of heavy metal exposure directly affecting the brain is lacking. While research on manganese exposure in the agricultural context did not find an association with prefrontal activity, previous findings showed that manganese-exposed welders demonstrated altered prefrontal cortex activity when performing a cognitive task [175,176]. This suggests the importance of focusing specifically on agricultural exposure as there may be important differences from other occupations. One review of important risk factors for psychiatric disorders for farmers in Brazil considered exposure to nicotine to be among the most important risk factors but only two articles were found that that qualified for this review concerning nicotine exposure and the brain [167].

## 5. Conclusions

In systematically reviewing the literature, we identified four challenges that hinder our development of a clear and comprehensive understanding of agricultural risks for brain health. First, there is great diversity in agricultural work. A large dairy manager, for example, is likely to have a much different experience in terms of chemical, environmental and mental exposures than a dairy production operator [142]. Similarly, the environmental risks encountered by a large acreage wheat farmer, may be much different from a small acreage organic avocado farmer and possible protective factors such as organic farming must be explored [20,177]. In order to better understand the independent, cumulative, and multiplicative impact of agricultural risks, better measures and classification of these occupation subtypes and the duration of time in that profession are needed. Second, and specific to the study of brain health, we found only three human studies that included direct imaging measures of human brain structure and function in their research. Although animal studies and epidemiological studies of clinical disease and disorder can provide important clues as to the specific types of brain abnormalities caused by risks, brain imaging and/or other direct measures of brain structure and function are needed to clarify assumptions and identify yet unknown mechanisms of injury and disease.

Furthermore, there has been overwhelmingly large focus on pesticides and chemical exposures in this line of research. We were surprised to find very little research on the brain-related impacts of risk factors that are known to contribute to cognitive and emotional health such as work-related stress, cognitive demand on the job, and likelihood of injury during production. The prevalence of mental health disorders, including depression and suicide, among agricultural workers is higher than in the rest of the population, but very little work has been done to understand the neurological mechanisms underlying this trend [15,178,179].

Other physical risk factors have been observed in agricultural occupations such as exposure to sun, vibration, extreme temperatures, noise, and long working hours but none of these have been explored for impacts on the brain. Importantly, when investigating these relatively untouched areas, moderators such as race, nationality, age, and sex must be accounted for. Finally, little work is currently being done to identify the possible factors related to agricultural work that could have protective effects in the brain despite some studies that suggest benefits associated with farm work [124]. Despite a trend in new research investigating “agricultural therapy” and the prosocial effects of farm communities, we did not find research that considered protective factors for brain health in agricultural workers, such as social engagement, family support systems, or dietary/lifestyle factors [180,181,182].

As noted above, future studies need to examine the impact of specific agricultural risk factors and brain health using direct measures of brain structure and function. Since agriculture employs more workers over the age of 55 than any other industry, literature also needs to consider the impacts these risk factors may have on brain aging [3]. Previous research has found a link between occupational physical stress and hippocampal volume in older adults as well as beneficial impacts of farm-work on dependency in old age [183,184]. Developing a better understating of these factors may one day improve our ability to develop recommendations that can protect the health of this absolutely vital sector of the work industry.

## Figures and Tables

**Figure 1 ijerph-19-03373-f001:**
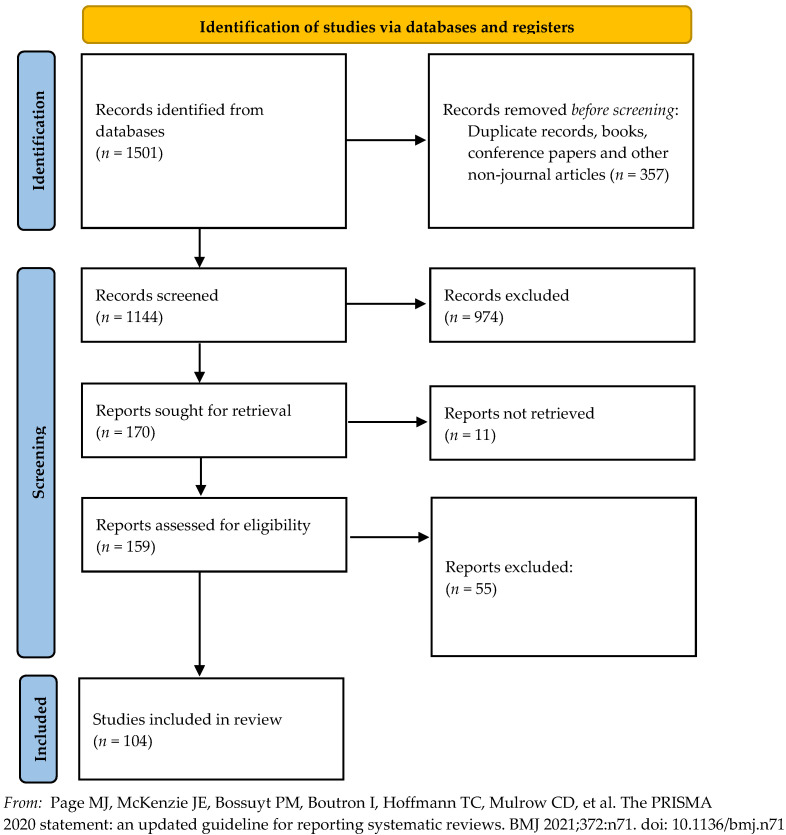
PRISMA diagram of search and review process.

**Figure 2 ijerph-19-03373-f002:**
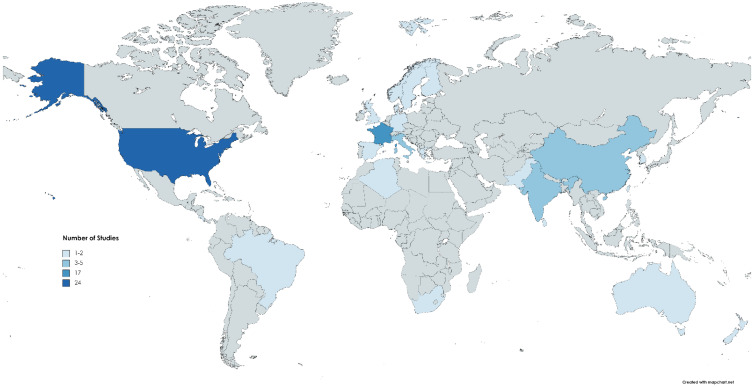
Number of agricultural workers’ brain health studies published by country not including animal studies or international studies.

**Table 1 ijerph-19-03373-t001:** Articles included in the review sorted by risk factor, brain area, and type of study with notes on participant demographics.

Number of Studies	Risk Factor	Relevance to the Brain	Study Types	Findings	Demographics
33	Non-Specific Factors Associated with Agricultural Work/Farming [12,21,25,48,49,50,51,52,53,54,55,56,57,58,59,60,61,62,63,64,65,66,67,68,69,70,71,72,73,74,75,76]	Brain Cancer [49,51,55,57,58,61,64,66,67,74] Parkinson’s Disease [21,25,48,50,52,53,54,56,59,60,62,63,65,68,69,70,71,72,73,75,76] Alzheimer’s Disease [25] Dementia [12]	Case-Control [48,49,51,53,54,55,56,57,58,59,60,61,63,64,65,66,67,68,69,71,72,73,75,76] Longitudinal [12,21,52,62] Cohort [25,50,74]	Agricultural occupations are generally associated with increased risk for brain cancer, Parkinson’s Disease, and Alzheimer’s. Disagreements in studies could be related to lack of information about duration of occupation, or lack of contending for important moderating factors like sex, race, age, and type of work.	Usually males and females ages 20–85, as well as mostly White participants.
1	Airborne Toluene [77]	GABAergic Neurons/Pathways [77] Cholinergic [77] Neurons/Pathways [77]	Animal [77]	Airborne toluene was associated with changes in the fluorescence intensity and morphology of GABAergic and cholinergic neurons in *C*. *elegans*.	NA
1	Dust [78]	1 Glial Cells [78]	Animal [78]	Organic dust from an agricultural work site activated HMGB1-RAGE signaling axis in *C. elegans* to induce a neuroinflammatory response in glial cells.	NA
5	Farm Animals Exposure [79,80,81,82,83]	Brain Cancer [79,80] Multiple Sclerosis [81,82] Parkinson’s Disease [83]	Case-Control [79,80,81,82,83]	Results were mixed with three studies finding associations between exposure to farm animals and adverse brain-health effects, one of which found the association only for women, and two studies did not find an association.	Usually males and females aged 20–60. One study included adults 60–80.
6	Heavy Metals [84,85,86,87,88,89]	Dorsolateral Prefrontal Cortex [84] Cholinergic Neurons/Pathways [85] Dopaminergic Neurons/Pathways [85,87] Hippocampus [88,89] Parkinson’s Disease [86] Striatum [87]	Human Imaging [84] Animal [85,87,88,89] Epidemiological [86]	Five of the six articles found an association between heavy metal exposure and adverse brain impacts.	Usually males only due to availability of data and participants.
2	Nicotine Exposure [90,91]	Dorsolateral Prefrontal Cortex [90] Putamen [90] Cerebellum [90] Functional Networks [91]	Neuroimaging [90,91]	Farmworkers exposed to nicotine plants had greater gray matter signal in putamen and cerebellum and lower gray matter signal in frontal and temporal lobes and differences in functional networks associated with biomarkers of nicotine exposure.	Latino males ages 30–70.
57	Pesticides	Amygdala [92,93] Antioxidant enzyme activity [94] Axonal transport [95] Brain Mitochondria [96] Brain Cancer [97,98,99,100,101,102] Cholinergic Neurons/Pathways [103] Cerebellum [90,104] Cerebral Cortex [104,105,106,107,108] Dopaminergic Neurons/Pathways [109,110] Functional Networks [91] Glutaminergic Neurons/Pathways [111] Glial Cells [111] Hippocampus [92,93,105,111] Neuronal Tubulin [112] Oxidative Stress [113,114] Parkinson’s Disease [25,50,54,60,63,83,114,115,116,117,118,119,120,121,122,123,124,125,126,127,128,129,130,131,132] Pineal Gland [133] Somatosensory Cortex [111,134,135] Striatum [105]	Animal [92,93,94,95,96,103,104,105,106,107,108,109,110,111,112,113,127,134] Case-Control [60,63,83,97,100,101,115,116,119,120,121,125,126,130,131,132,136,137,138] Cohort [50,54,98,99,122,124,128,129,132] Cross-Sectional [117,139] Ecological [118] Human Imaging [90,91] Longitudinal [133,135] Molecular Cellular [102,114,123]	Pesticides are generally associated with negative impacts on brain health. Moderating factors like sex, race, duration of exposure, and migrant status may be important considerations for future research.	The age groups studied in this area tend to be slightly older with most average ages in the 40s to 60s.

**Table 2 ijerph-19-03373-t002:** A representation of the risk factors and relevance to the brain with darker gray representing more articles (darkest gray = 23 articles), lighter gray representing fewer articles, and white representing no articles.

Relevance to the Brain	Agricultural Occupations	Airborne Toluene	Dust	Farm Animals	Heavy Metals	Nicotine Exposure	Pesticides
Brain Cancer							
Alzheimer’s Disease							
Parkinson’s Disease (Nigrostriatal Region)							
Dementia							
GABAergic Neurons/Pathways							
Cholinergic Neurons/Pathways							
Glutaminergic Neurons/Pathways							
Dopaminergic Cells/Pathways							
Neuronal Tubulin							
Antioxidant enzyme activity							
Glial Cells							
Somatosensory Cortex							
Amygdala							
Pineal Gland							
Putamen							
Hippocampus							
Striatum							
Cerebellum							
Multiple Sclerosis							
Dorsolateral Prefrontal Cortex							
Functional Networks							
Oxidative Stress							
Cerebral Cortex							

## Data Availability

Not applicable.

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
