# Peer review of "Risk Factors for Brain Health in Agricultural Work: A Systematic Review"

_ijerph, 2022, doi:10.3390/ijerph19063373_

Round 1

Reviewer 1 Report

The topic taken up by the authors is of great importance for public health, especially in the agricultural sector. The authors have conducted a thorough literature review and summarised the reports in a constructive manner. The work presented to me for review entitled "Risk Factors for Brain Health in Agricultural Work: A Systematic Review" is prepared in a sound manner and with great commitment, and in my opinion is ready for publication in the presented form.

However, there are some minor editorial errors such as the description of figures, which should be placed under the figure.

Additionally, Figure 2 is a table rather than a figure.

In the "materials and methods" section, lines 160-165 do not seem to be relevant to the content.

Having these minor errors corrected, the article is suitable for publication.

Author Response

REVIEWER 1

The topic taken up by the authors is of great importance for public health, especially in the agricultural sector. The authors have conducted a thorough literature review and summarised the reports in a constructive manner. The work presented to me for review entitled "Risk Factors for Brain Health in Agricultural Work: A Systematic Review" is prepared in a sound manner and with great commitment, and in my opinion is ready for publication in the presented form.

However, there are some minor editorial errors such as the description of figures, which should be placed under the figure.

We appreciate the reviewer noting this editorial error. Figure descriptions are now placed under figures.

Additionally, Figure 2 is a table rather than a figure.

Figure 2 has been changed to Table 2.

In the "materials and methods" section, lines 160-165 do not seem to be relevant to the content.

Lines 160-165 (now lines 171-173 on page 4) provide a narrative for Figure 1, a PRISMA diagram that depicts the flow of information through the different phases of a systematic review.

Having these minor errors corrected, the article is suitable for publication.

We appreiate the supportive comment.

Reviewer 2 Report

This is a well conducted and rigorous literature analysis of studies focused on brain health in agricultural occupations. Very useful information is provided about the different risks and the impacted brain areas. Overall, the manuscript is already well written. Some points are a bit confusing: the inclusion of laboratory studies for non agricultural-specific exposure such as metals, or dust, is misleading. The authors indicate that these studies were included if they were specifically related to an 'agricultural context'. It is difficult to understand how an animal study on manganese exposure, for example, can be attributed to an agricultural context.

There is also a lack of consideration about race and ethnicity. Migrant and immigrant workers represent the majority of agricultural workers but are highly understudied, as this study also shows that most of the research is about white workers. Therefore, exposure to agricultural risk factors may be underestimated because not including the most exposed workforce.

There is also a lack of consideration of mixed exposure that may impact the brain more consistently than the individual risk factors. If no studies were located on mixed exposure in agriculture as potential risk for brain health, this should be underlined as a gap to be addressed in future studies.

  • line 620 repeated word ' examine-examine'
  • line 153: what is the 'agricultural context'?

Author Response

REVIEWER 2

This is a well conducted and rigorous literature analysis of studies focused on brain health in agricultural occupations. Very useful information is provided about the different risks and the impacted brain areas. Overall, the manuscript is already well written. Some points are a bit confusing: the inclusion of laboratory studies for non agricultural-specific exposure such as metals, or dust, is misleading. The authors indicate that these studies were included if they were specifically related to an 'agricultural context'. It is difficult to understand how an animal study on manganese exposure, for example, can be attributed to an agricultural context.

We appreciate the reviewer’s comment. To clarify we have amended the methods on page 3-4 lines 149-156 to include further explanation and an example of how the agricultural context was determined. Specifically, we write, “For example, a study using mouse microglial cells to investigate the neuroinflammatory effects of organic dust was included because the dust used in the study was collected from a farm where the risk of inhalation is present for agricultural workers thus making the research directly relevant to occupational risk factors [48]. However, multiple animal studies were excluded, despite focusing on well known risk factors in agriculture, like pesticide exposure, due to a lack of explanation of how their methods specifically re-late to agricultural work.” Additionally, we have added more detail on the direct relevance of the animal research on heavy metals on page 4 lines 369-371. Specifically, we write, “The authors also investigated whey protein isolate as a protecting factor against these effects as part of their aim to improve outcomes from occupational exposure to heavy metals [85]”. We also changed lines 374-377, “Importantly, this study observed rats after low dose, subchronic pesticide exposure far lower than what would usually be considered acute poisoning and more similar to an agricultural worker who may handle pesticides safely enough to experience only low dose exposure [87]”

There is also a lack of consideration about race and ethnicity. Migrant and immigrant workers represent the majority of agricultural workers but are highly understudied, as this study also shows that most of the research is about white workers. Therefore, exposure to agricultural risk factors may be underestimated because not including the most exposed workforce.

We thank the reviewer for raising this important point. We have added discussion of this issue on p. 8 lines 544-546. Specifically, we now write, “Given that migrant workers make up the majority of the agricultural workers, including migrant workers in agricultural research will help fill critical knowledge gaps for an increasingly important segment of agriculturalists.”

There is also a lack of consideration of mixed exposure that may impact the brain more consistently than the individual risk factors. If no studies were located on mixed exposure in agriculture as potential risk for brain health, this should be underlined as a gap to be addressed in future studies.

This is another great point. We have added to the discussion, page 7 lines 514-515, an explanation that mixed exposures are indeed a gap in the literature.  “Additionally, mixed exposures are rarely studied but represent an important line of research in brain health.” We also now write on page 8 lines 564-572 “Despite the fact that agriculturalists likely face mixed exposures to the risk factors reviewed here, mixed exposures are rarely studied. In an exception, one recent article found that the combined effects of corticosterone, an indicator of physiological stress, and chlorpyrifos, a common pesticide, was related to neuroinflammation and effects on neural signaling. More studies like this are needed in we hope to fully understand risks for brain health in this population.”

Line 620 repeated word ' examine-examine'

We have fixed this typo.

Line 153: what is the 'agricultural context'?

We appreciate the opportunity to clarify our meaning. We have replaced this term with a detailed description as part of our response to reviewer 2’s first comment. On page 5 lines 149-156, we now write, “Animal studies were included if they discussed implications for agriculture. For example, a study using mouse microglial cells to investigate the neuroinflammatory effects of organic dust was included because the dust was collected from a farm where the risk of inhalation is present for agricultural workers thus making the research directly relevant to occupational risk factors [48]. However, multiple animal studies were excluded, despite focusing on well known risk factors in agriculture, like pesticide exposure, due to a lack of explanation of how their methods specifically relate to agricultural work.”

Reviewer 3 Report

Reviewer’s comments

Sturm et al have reviewed the potential risk factors for neurological disorders occurs in various type of agriculture-related workers. In this review, they described risk factors, sources, affected brain areas, neurological disturbances, sex specific effects in animal and human studies. Their literature review data were very interesting, well-written and would be helpful not only for environmental, occupational, and public health related researchers and but also for authorities of environmental conservation and occupational health.

  • In abstract, the authors stated that nigral striatal regions were the most well studied brain area impacted. This result represents all neurological disorders, or it depends on the type of chemical studied? How do you think?
  • Human subjects could not manipulate as animal studies and recent non-invasive neuroimaging techniques may be helpful to assess the changes in brain morphology including volume of gray matter, neurotransmitters and metabolites and specific area abnormalities. However, this neuroimaging facilities may be available in developed countries and not in developing countries. Most of the developing countries, especially in rural areas, agriculture is the main work. Thus, socioeconomic and education status are important for prevention of exposure to chemicals from agricultural work (e.g., proper usage and handling of pesticides, knowledge of health risk and personal hygiene to prevent the exposure to self and family members especially risk persons like pregnant women, children and elderly persons) and this issue should be discussed.
  • Developmental exposure to chemicals used in agricultural processes such as pesticides may lead to neurodevelopmental disorders like ASD, ADHD etc., Thus, it should be added under 3.1. Non-Specifical Factors Associated with Agricultural work.
  • In page 4, line 246: The authors stated that higher socio-economic status was associated with higher incidence of brain cancer, what does it mean?
  • How do you suggest the possible reasons of sex-specific effects in brain diseases in agricultural work.
  • Personal hygiene, lifestyle and proper handling of toxic chemicals including pesticide are also important for worker and their families.
  • Are there any literature showing the intervention to eliminate source or cause and any benefit after intervention?
  • Prevention is better than cure. Based on literature review, toxicity testing method, regular screening for early diagnosis and termination or substitution of dangerous toxic chemicals and providing health education to public etc., are critical role in the reduction of neurodevelopmental disorders in future generation. This issue should be discussed.  

Author Response

REVIEWER 3

Sturm et al have reviewed the potential risk factors for neurological disorders occurs in various type of agriculture-related workers. In this review, they described risk factors, sources, affected brain areas, neurological disturbances, sex specific effects in animal and human studies. Their literature review data were very interesting, well-written and would be helpful not only for environmental, occupational, and public health related researchers and but also for authorities of environmental conservation and occupational health.

We appreciate the reviewer's supportive comment.

In abstract, the authors stated that nigral striatal regions were the most well studied brain area impacted. This result represents all neurological disorders, or it depends on the type of chemical studied? How do you think?

This is a good question. We have added the following comment to p. 9 lines 587-592, “Additionally, the nigrostriatal dopaminergic system was the most commonly studied brain region due to its well-described vulnerability to pesticides and important impli-cations with Parkinson’s disease and other movement disorders. Expanding under-standing of other brain areas impacted by pesticides will help aid in future efforts to identify early signs of disease and disorder as well as contribute to protective measures.”

Human subjects could not manipulate as animal studies and recent non-invasive neuroimaging techniques may be helpful to assess the changes in brain morphology including volume of gray matter, neurotransmitters and metabolites and specific area abnormalities. However, this neuroimaging facilities may be available in developed countries and not in developing countries. Most of the developing countries, especially in rural areas, agriculture is the main work. Thus, socioeconomic and education status are important for prevention of exposure to chemicals from agricultural work (e.g., proper usage and handling of pesticides, knowledge of health risk and personal hygiene to prevent the exposure to self and family members especially risk persons like pregnant women, children and elderly persons) and this issue should be discussed.

The reviewer makes a strong point about socio-economic status. We have addeded the following comment to page 8 lines 549-563, “One study that investigated trends in socio-economic status associated with Parkinson’s disease rates found that farmers had increased risk but socio-econimic status had a relatively small effect on the liklihood of hospitalization for Parkinson’s disease [1]. Suprisingly, another study found that higher socio-economic status was linked with increased rates of brain cancer, although the same study did not find elevated risks for farmers like most other epidemiological studies [2]. It’s possible that using death certificates as a measure of occupation inhibits the ability to fully explore these effects since the occupation coded is “usual” occupation and may miss occupational exposures from other than usual occupation. Methodological challenges also impact the ability to study socio-economic status. For example, brain-imaging equipment is typically located in more urban areas where it is difficult to recruit agriculturalists for research. Additionally, farmers’ lifestyle’s can often leave them with little flexibility in free-time and scheduling that would be needed to participate in a lengthy research study [3]. Future research should make a stronger efforts to take socio-economic status, education, and cultural barriars to research participation into consideration.”

Developmental exposure to chemicals used in agricultural processes such as pesticides may lead to neurodevelopmental disorders like ASD, ADHD etc., Thus, it should be added under 3.1. Non-Specifical Factors Associated with Agricultural work.

Thank you for pointing this out. Due to the existence of previous reviews focused on developmental risk factors, we focused on adult populations in the paper. We have added the following comment to p. 3 lines 141-143, “Developmental studies were not considered due to the existence of other reviews and meta-analyses that focus on neurodevelopment in relation to agricultural risk factors [17,42–47]. However, understanding risks throughout the lifetime will be important to the broader scope of recommendations aimed at reducing harm.”

In page 4, line 246: The authors stated that higher socio-economic status was associated with higher incidence of brain cancer, what does it mean?

Surprisingly, one study did find an association between higher socio-economic status and increased rates of brain cancer. I have added the language below to page 8 lines 553-557 to expand on this point.

“Suprisingly, another study found that higher socio-economic status was linked with increased rates of brain cancer, although the same study did not find elevated risks for farmers as did most other epidemiological studies [2]. It’s possible that using death certificates as a measure of occupation limits the ability to fully explore these effects.”

How do you suggest the possible reasons of sex-specific effects in brain diseases in agricultural work.

This is interesting topic, but it also challenging to address because the unbalanced research. We have added the following comment to pages 7-8 lines 529-544, “Sex-specific effects cannot be determined from much of the research included in this review due to the recruitment of mostly male participants. While the industry is overwhelming staffed by male workers (68% in 2016), females are increasingly working in agriculture (down from 72% men in 2014) [160]. (Alavanja et al., 1996; The Sexual Division of Farm Household Labor, 1988). Also, while principal operators are more likely to be male, secondary operators are more likely to be female [161]. Studies that did include females found increased rates of Parkinson’s disease among male farmers and increased rates of dementia and brain cancer among women farmers  [21,53,67,74,119]. Additionally, females who perform “off-farm” labor may also be exposed to many of the same risks of agricultural work even though they are not counted among the employed agricultural population [162]. Importantly, men may be more likely to experience injury and pesticide exposure on the job [163]. It is likely that these findings, overall, reflect some combination of differences in sex defined as a biological variable as well as also socio-cultural factors related to gender norms in the work enviorment. As gender norms change, risk factors and affected groups will also likely change with them.”

Personal hygiene, lifestyle and proper handling of toxic chemicals including pesticide are also important for worker and their families.

This review did not address sources of exposure therefore, while these are imporant points they are beyond the scope of the review.

Are there any literature showing the intervention to eliminate source or cause and any benefit after intervention?

Intervnetions was not the purpose of the review we conducted.

Prevention is better than cure. Based on literature review, toxicity testing method, regular screening for early diagnosis and termination or substitution of dangerous toxic chemicals and providing health education to public etc., are critical role in the reduction of neurodevelopmental disorders in future generation. This issue should be discussed.

These are important points but our review was designed to assess what is known about brain health in agricultural populations and we did not address interventions and prevention strategies, which have been discussed extensively by others.